# Exposing telomere length's impact on malnutrition risk among older adults residing in the community: Insights from cross-sectional data analysis

Priscila Rodrigues[1], Guilherme Furtado[2,3,4]*, Margarida Martins[1,5,6,7,8,9,10], Ricardo Vieira[11], Ariene Orlandi[12], Sónia Brito-Costa[2,13]*, Ana Moisão[2,13], Ligiana Corona[14], Daniela Lima[1], Tábatta Brito[1]*

1 Faculty of Nutrition, Federal University of Alfenas, Alfenas, Brazil, 2 Polytechnic University of Coimbra, Rua da Misericórdia, Lagar dos Cortiços–S. Martinho do Bispo, Coimbra, Portugal, 3 Research Centre for Natural Resources Environment and Society (CERNAS), Polytechnic University of Coimbra, Coimbra, Portugal, 4 SPRINT - Sport Physical activity and health Research & INnovation cenTer, Polytechnic University of Coimbra, Coimbra, Portugal, 5 GreenUPorto - Sustainable Agrifood Production Research Centre, Vairão, Portugal, 6 Centre for the Research and Technology of Agro-Environmental and Biological Sciences (CITAB), Vila Real, Portugal, 7 Polytechnic University of Coimbra, Coimbra, Portugal, 8 H&TRC - Health & Technology Research Center, Coimbra Health School, Polytechnic University of Coimbra, Coimbra, Portugal, 9 Sports and Physical Activity Research Center, University of Coimbra, Coimbra, Portugal, 10 Research Centre for Anthropology and Health, University of Coimbra, Coimbra, Portugal, 11 Nursing School, Federal University of Alfenas, Alfenas, Brazil, 12 Nursing Department, Federal University of São Carlos, São Carlos, Brazil, 13 InED - Center for Research and Innovation in Education, Polytechnic Institute of Porto, Porto, Portugal, 14 Faculty of Applied Sciences, University of Campinas, Campinas, Brazil

* sonya.b.costa@gmail.com, sonia.costa@ipc.pt (SBC); tabatta.brito@unifal-mg.edu.br (TB); guilhermefurtado@ipc.pt (GF)

## Abstract

### Background

Successful aging is associated with an increase in life expectancy. For a better understanding of the aging process, recognize the relationship between telomere length and nutritional status is a novel approach in geriatric science. Telomers shortening coincides with a decrease in life expectancy, and an increased risk of malnutrition-related diseases.

### Goals

The goal of this study was to investigate whether a shorter telomere length is associated with a greater likelihood of malnutrition in community-dwelling older adults.

### Methods

A cross-sectional study with a probabilistic sample of 448 older people aged 60 years old or over, and living in the urban area of an inland Brazilian municipality was conducted. The information was gathered in two stages: a) a personal interview was conducted to obtain sociodemographic, cognitive, and functional autonomy data. The Mini Nutritional Assessment was used to assess the risk of malnutrition. b) a blood sample was taken to proceed

**Data Availability Statement:** The data underlying the findings include potentially identifying

participant information and cannot be shared publicly due to ethical/legal restrictions. However, researchers interested in collaborating and accessing the data should contact the Faculty of Nutrition, University of Alfenas, via email (fanut@unifal-mg.edu.br).

**Funding:** Tábatta Brito was funded by the Fundação de Amparo à Pesquisa do Estado de Minas Gerais (FAPEMIG) (Grant No. APQ-01168-18; 001/2018). Daniela Lima was funded by the National Council for Scientific and Technological Development (CNPq) (Grant No. 429823/2018-5-MCTIC/CNPq No. 28/2018). Guilherme Furtado acknowledges support from the Foundation for Science and Technology (FCT) under the institutional scientific employment program-contract (CEECINST/00077/2021). Sonia Brito-Costa and Ana Moisão acknowledges support from the Fundação para a Ciência e Tecnologia (FCT) under the scope of the project UIDB/05198/2020 (Centre for Research and Innovation in Education, inED; Link: https://doi.org/10.54499/UIDB/05198/2020). The authors thank all the funders for their support. The funders had no role in study design, data collection and analysis, decision to publish, or preparation of the manuscript.

**Competing interests:** The authors have declared that no competing interests exist.

with the relative quantitative study of telomere length using real-time qPCR method. The differences between the groups were estimated using Pearson's v2 and Fisher's exact tests. In the data analysis, descriptive statistics and multiple logistic regression were applied.

## Results

In 34.15% of the total sample, malnutrition was recognized as a risk factor. Older people with the shortest telomere length had more chances of getting malnutrition (OR = 1.63; IC:95% = 1.04–2.55) compared to those with longer telomeres, independent of age groups, family income, multimorbidity, cognitive decline, and depressive symptoms.

## Conclusion

The creation of clinical trials and the implementation of therapies to reduce the risk of malnutrition will be aided using the telomere length as an aging innovative biomarker, connected with nutritional status.

## 1. Introduction

Population aging is a global reality that accompanies the increase in longevity. This brings forth challenges to be overcome in biological, social, and cultural aspects, as the rise in life expectancy is accompanied by the pursuit of healthy aging [1, 2]. However, chronological aging differs from biological aging. The former is determined by the number of years on the calendar, while the latter reflects the age of the organism, indicating its level of functional decline, which can be measured through aging biomarkers [3, 4].

Biological markers are used as experimental parameters to measure and indicate the occurrence of specific normal or pathological functions in the organism [5]. Among potential biomarkers of human aging, telomere length has been proposed as a promising marker [6]. Telomeres are non-coding DNA nucleoprotein structures located at the ends of eukaryotic chromosomes [7]. It is believed that telomere length gradually decreases throughout life, making it a possible biomarker of aging and an important predictor of age-related diseases [8].

In the context of the aging process, biomarkers play a crucial role in comprehending the process and facilitating the development of essential interventions aimed at reducing morbidity and mortality [4]. These include physiological markers (e.g., blood pressure), physical markers (e.g., waist circumference), biological regulators (e.g., interleukin-6), cellular markers (e.g., lipoproteins), genetic markers (e.g., telomeres), enzymatic markers (e.g., telomerase), hormonal markers (e.g., leptin), among others [4, 5, 9]. Several biological markers have been studied to identify diseases and measure the therapeutic effects of drugs. However new genes, proteins, and metabolites can be used to study the biology of aging [4, 10, 11].

One of the current focal points in studying human aging lies in the investigation of telomere length as a primary biomarker [6]. Telomeres, situated at the termini of eukaryotic chromosomes, are non-coding DNA nucleoprotein structures [12]. Comprised of multiple repetitions of a specific DNA base sequence (TTAGGG) along with associated proteins, their vital role involves safeguarding the DNA ends of the chromosome and impeding the loss of genetic material [7].

The shortening of telomeres, segments of chromosomes, over multiple cell divisions lead to cellular senescence, where cells lose the ability to divide, repair damage, and undergo apoptosis [13, 14]. This state of replicative senescence is a potential factor in human aging, along with

other elements that compromise the efficiency of cellular repair processes [15]. Studies suggest that telomere shortening can serve as a potential biomarker to determine cell proliferation capacity and indicate the state of senescence [6, 8, 16]. Moreover, nutritional status can be defined as the extent to which physiological requirements for nutrients are fulfilled to uphold appropriate body composition and function [17], thereby serving as a significant indicator of overall health among older individuals [18].

Several factors are associated with nutritional status in older individuals, including health conditions and comorbidities [19] socioeconomic status [20, 21], level of education [22], lifestyle behavior [23], as well as environmental factors (e.g., availability of fresh, healthy and varied foods) [24]. Recent studies have taken a more integrated approach, considering factors such as the importance of physical exercise associated with adequate protein [25], and vitamin (especially d) intake [26], dental health related to the ability to consume solid foods [27], optimal functioning of the microbiota linked to micronutrient absorption capacity [28], and preservation of cognitive abilities as gold-standard markers for active and healthy aging [29].

Despite the advancements in nutritional science aimed at understanding the aging process, there has been a substantial increase in studies examining the relationship between biomarkers and nutritional status in recent years [14]. A systematic review and meta-analysis conducted by [30] identified 43 blood biomarkers that were associated with the risk of malnutrition in older adults, including notable markers such as albumin, hemoglobin, total cholesterol, and total lymphocyte count [30]. Furthermore, another study recently conducted demonstrated that malnutrition risk in older individuals was linked to different factors such as the presence of heart valve disease, a high number of diseases, and female sex. Elevated levels of inflammatory biomarkers, specifically logsTNFRII, IL-8, and OPG, were associated with an increased risk of malnutrition, while higher levels of IL-18 were associated with a reduced risk. These findings indicate that chronic inflammation and immunologic processes play significant roles in the intricate development of malnutrition in older individuals [31].

In addition to blood biomarkers, there is a growing interest in investigating the relationship between genetic biomarkers and nutritional status [30, 32–34] Studies have explored the role of telomeres, which are genetic structures related to DNA stability and replication, as potential biomarkers of aging and nutrition-related diseases [8]. A study aimed at exploring the correlation between telomere length and prenatal famine revealed that such exposure was not significantly associated with telomere length. The remaining studies predominantly focus on investigating the consequences of prenatal and early-life malnutrition on adulthood and the aging process [32]. One study involving 1,859 participants (mean age 63.6 years) concluding that exposure to prenatal famine was associated with the risk of late-life depression, with telomere length partially mediating this relationship [33].

Considering that nutritional status and telomere length are important issues related to aging, exploring their relationship, which may generate stronger evidence for future prevention and treatment interventions for malnutrition, will contribute to strengthening the hypothesis that telomeres are significant biomarkers. Therefore, this study aims to answer the following research question: do older adults with shorter telomere length have a higher risk of malnutrition? It is believed that older adults with shorter telomeres are more likely to be at risk of malnutrition and/or malnourishment.

## 2. Methods

### 2.1 Study design

This is a subset of the project entitled "Association between low level of social support and telomere length in older adults" [35]. It is a quantitative study with an analytical cross-sectional

design. The structure of the present study followed the guidelines of the STROBE (Strengthening the Reporting of Observational Studies in Epidemiology) initiative [36].

## 2.2 Settings

The study was conducted in the municipality of Alfenas, located in the southern region of the state of Minas Gerais, Brazil. According to projections from the Brazilian Institute of Geography and Statistics (IBGE), the population of Alfenas in 2019 was 79,996 inhabitants. At the time of sample calculation, the latest available age-specific population projection was provided by the Interagency Network for Health Information (RIPSA) for the year 2015, which indicated a population of 78,713 inhabitants, with 10,797 being older adults (Instituto Brasileiro de Geografia e Estatística, 2020).

## 2.3 Participants

The study population consisted of community-dwelling older individuals aged 60 years and older, residing in the urban area of the municipality of Alfenas (in 2019). The sample size calculation was obtained considering the estimation of proportions around 50%, a 95% confidence interval, a design effect of 1.17, and a population of 10,797 older people, resulting in a sample of 435 individuals. The final sample size (n = 448) is depicted in Fig 1 and considered the addition of extra individuals to compensate for potential losses.

## 2.4 Sample selection criteria

The inclusion criteria were being 60 years of age or older and having the ability to respond to the questionnaire (as perceived by the interviewer during the presentation of the research and invitation to participate). The exclusion criterion was having permanent or temporary incapacity to walk, except with the use of a walking aid device.

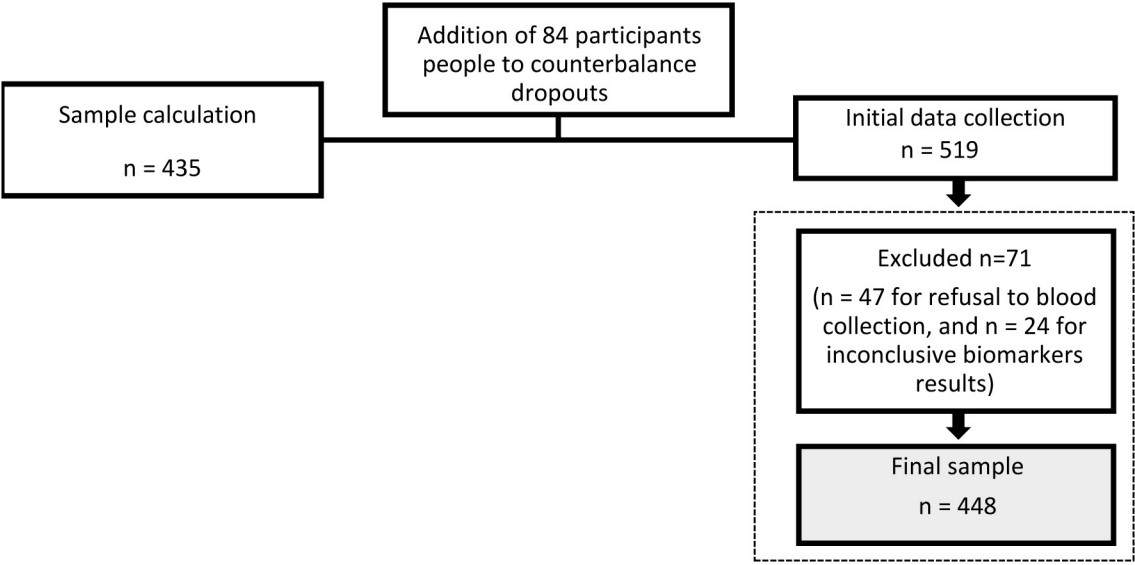

**Fig 1. Flowchart of participants recruitment.**

## 2.5 Data collection procedures

The participants were recruited from selected households to ensure that individuals from all regions of the municipality were included. The sampling process used as a basis was the complementation of the sample conducted in the SABE Study, a population-based survey conducted in the city of São Paulo. The interviewers were distributed in different regions of the city according to the proximity to their place of residence. After locating a household with a resident aged 60 years or older, they proceeded to locate nearby households, or at most, within the boundaries of the neighborhood to which the initial address belonged [37].

Data was collected between July and December 2019 in two phases. In the first phase, interviews and physical-functional assessments were conducted. A comprehensive survey was developed, including questions from the following sections: a) sociodemographic characteristics; mental wellbeing and social support; c) general health status; d) functionality, sarcopenia, frailty, and falls; nutritional status; f) physical assessment. In the second phase, fasting blood samples were collected. The interviews took place at the older person's residence, while the blood sample collection was performed at the Central Laboratory of Clinical Analysis (LACEN) of the Federal University of Alfenas (UNIFAL-MG) or at the older person's residence for a registered nursing.

## 2.6 Ethics

This study was submitted to the Research Ethics Committee of the Federal University of Alfenas and received approval under CAAE: 5218518.0.0000.5142 [No. 2.668.936]. Older adults were included in the study after providing informed consent by signing the informed consent form. All procedures were conducted in accordance with the ethical standards outlined in the Declaration of Helsinki [38], and following the ethical precepts recommended by Resolution 466/12 of the National Health Council for experiments involving humans [39].

## 3. Measurements

### 3.1 Telomere length assessment

Initially, a sample of whole blood was obtained in tubes containing EDTA and transported in a polystyrene box with reusable ice to the Clinical Parasitology Laboratory of the Federal University of Alfenas. The blood samples were processed using the basic DNA extraction protocol with affinity columns. After adding a saline phosphate buffer (pH 7.2) and centrifuging at 2.000 rpm for 5 minutes, the supernatant was discarded. The remaining precipitate was treated with an erythrocyte lysis buffer (ACK) until complete hemolysis. The material was centrifuged again (2,000 rpm for 5 minutes), and the supernatant was discarded. The remaining precipitate was treated with a specific buffer from the extraction kit (PureLink™ Genomic DNA Mini Kit, Invitrogen) and incubated in a water bath at 56°C. The DNA extraction was performed according to the manufacturer's recommendations, cited um previous analogous researchers [40]. The DNA concentration was determined using a Genova Nano spectrophotometer (Jenway), and the concentration was adjusted to 10ng/μl in the samples, which were then stored at -20°C until the PCR (Polymerase Chain Reaction) reactions were performed [41].

Real-time PCR reactions were conducted using Power SYBR Green PCR Master Mix as the fluorescent marker in an ABI StepOne Real-Time PCR System (Applied Biosystems), with a final volume of 10μL per reaction. A volume of 1μL of DNA (samples or controls) was added to a mix containing 5 μL of 2X Power SYBR® Green Master Mix, 1μL of a mixture including the forward and reverse primers (concentration of 10μM), and 3μL of ultrapure water, free of DNAse and RNAse. Negative and positive controls were included in the reactions. The

amplifications for telomere markers consisted of an initial cycle at 95˚C for 10 minutes, followed by 26 cycles at 95˚C for 15 seconds, 57˚C for 30 seconds, and 60˚C for two minutes. Subsequently, a melting curve stage was performed, starting at 95˚C for 15 seconds, followed by 60˚C for one minute. The temperature was then incrementally increased, and the readings were taken after each 0.3˚C increment until reaching a final temperature of 95˚C for 15 minutes. For the human beta-globin markers, the amplification consisted of an initial cycle at 95˚C for 10 minutes, followed by 40 cycles at 95˚C for 15 seconds, 56˚C for 45 seconds, and 60˚C for one minute. The melting curve stage was then performed following the same procedure as described above.

For telomere identification, the markers described by Cawthon (2009) were utilized. The telomere sequence used in this study was "telg." 5′ACA CTA AGG TTT GGG TTT GGG TTT GGG TTT GGG TTA GTGT3′ and telc: 5′TGT TAG GTA TCC CTA TCC CTA TCC CTA TCC CTA TCC CTA ACA3′, and the markers for the control gene, human beta-globin, were hbgu: 5′CGG CGG CGG GCG GCG CGG GCT GGG CGG ctt cat cca cgt tca cct tg3′ and hbgd: 5′GCC CGG CCC GCC GCG CCC GTC CCG CCG gag gag aag tct gcc gtt3′ (Cawthon, 2009).

To obtain the average relative telomere length, a mathematical model based on the exponential ratio of the number of copies of 41 telomere sequences for each individual compared to the number of copies of the single-copy gene was used [42]. For the categorization of the variable in this study, the distribution of the average relative telomere length was determined by the median. Older adults with a relative telomere length falling within the top 50% of the sample, indicating a greater difference compared to the single-copy gene, were classified as having a "shorter telomere length."

## 3.2 Nutritional status

The Mini Nutritional Assessment (MNA) was used to assess nutritional risk and malnutrition [43]. Its purpose is to screen for the risk of developing malnutrition or detect it at an early stage in older adults. The MNA is composed by 18 questions divided into two parts. The first part is called nutritional screening, and each question is assigned a score. If the sum of the screening points is 12 or more, further evaluation is unnecessary as there is no risk of malnutrition. If the score is ≤ 11 points, there is a possibility of malnutrition, and further evaluation is required [44]. The second part is called Global Assessment, and each question is also assigned a score. At the end, the scores are summed, and the nutritional status of the older adult can be classified as normal (score >23.5 points), at risk of malnutrition (score between 17 and 23.5 points), or malnourished, scored <17 points. Considering the small number of older adults identified with malnutrition, in this study, the nutritional risk and malnourished categories were combined, resulting in the binary variable "risk of malnutrition/malnutrition" with the categories "no" and "yes" [45].

## 3.3 Sociodemographic

Sociodemographic information was collected, namely sex (male, female), age group (60–69 years old, 70–79 years old, 80 years old and older); number of years of education (≤4 years of education, >4 years of education); family income (>2 minimum wages, between 1 and 2 minimum wages, <1 minimum wage); number of people in the household (living alone, with one other person, with two or more people).

### 3.4 General illness status

The classification of illness/comorbidities, and polypharmacy were the indicators used to assess the general state of health/diseases. Multimorbidity is defined as the presence of two or more self-reported chronic diseases in an individual [46]. Polypharmacy, on the other hand, refers to the reported continuous use of five or more medications by an individual [47].

### 3.5 Functionality

The Brazilian version of Katz scale and the Lawton & Brody scale were used to assess different dimensions of functional ability in this study [48]. The Katz scale measures the performance of basic activities of daily living (BADL), which encompass self-care tasks such as using the bathroom, getting dressed, showering, mobility, continence, and eating [49]. This scale indicates the level of dependency, with older people who can perform all BADL tasks without assistance considered independent [50]. On the other hand, the Lawton & Brody scale evaluates the performance of instrumental activities of daily living (IADL), which are more complex tasks required for independent living within the community [51]. These tasks include using transportation, managing household chores, shopping, making phone calls, managing finances, and taking medication [48].

### 3.6 Mental health status

A Brazilian version of the Cognitive Abilities Screening Instrument - Short Form (CASI-S) [52], and Geriatric Depression Scale (GDS) were used [53]. The CASI-S is a concise assessment tool designed to screen cognitive changes in older people. It evaluates temporal orientation, verbal fluency, spontaneous recall with semantic suggestion (category), and recognition [54]. Due to its brief administration time (approximately five minutes) and ease of use compared to traditional tests, the CASI-S holds great value for cognitive screening in population research. It has a maximum score of 33 points, and a cutoff point of 23 is commonly employed to identify cognitive decline [55]. The GDS is an instrument used to identify the presence of depressive symptoms in older people [56]. It consists of 15 yes/no questions and is considered a screening tool for depressive symptoms. A score of six or higher indicates a positive screening result for depressive symptoms. The GDS has been widely used in research and clinical settings to assess depression in older adults [53].

### 3.7 Anthropometric measures

The standardized procedures described by Lohan and colleagues were followed for the collection of anthropometric data [57]. To assess body mass, a portable scale (Seca®, model 770, Germany) with a precision of 0.1 kilograms was used. For stature, a portable stadiometer (Seca Body meter®, model 208, Germany) with a precision of 0.1 centimeters was used. Body mass index (BMI) was calculated according to the standard formula (BMI = body mass/stature$^2$). The calf circumference was measured with the older person's leg bent at a 90-degree angle with the knee, at its most prominent point (WHO. World Health Organization, 1995) Meanwhile, the arm circumference was measured on the non-dominant arm, with an inelastic measuring tape at the midpoint between the acromion and olecranon [58]. It is worth noting that the cutoff points used for BMI, AC, and CC were adopted following MAN recommendations [43].

## 4. Statistical analysis

To calculate the sample, the G* power statistics was used [59]. The database was constructed using Microsoft Office Excel version 2019 (16.0), incorporating double-data entries to rectify

potential typos. Proportions were estimated for categorical variables and group differences were analyzed using Pearson's chi-square test and Fisher's exact test. The Pearson's chi-square test was chosen since it tests the independence of two or more independent samples. The Fisher's exact test was used alternatively when the frequencies were less than five. Multivariate logistic regression was used to test the association since the dependent variable is dichotomous in nature (nutritional risk: no/yes). The magnitude of association was estimated by the crude and adjusted odds ratio (OR). Variables with a p-value < 0.20 in the univariate analysis were included in the final model using the stepwise forward procedure. Non-significant variables were retained in the final model for adjustment. The goodness of fit for the binary logistic models employed in this study was investigated using the Hosmer and Lemeshow test. To evaluate the quality of the final model, the area under ROC curve (AUROC) was used. In the regression model, telomere length was treated as a dependent variable, the nutritional status was analyzed as an independent variable and sociodemographic general health status, cognitive profile, and functionality were treated as potential confounders (co-variates). A significance level of 5% was used in all analyses. These statistical methods were employed to explore and assess relationships and associations between variables in the study, providing insights into the research objectives and allowing for a comprehensive analysis of the data.

All analyzes were performed using Stata® software version 17.0 (STATACORP LLC, 2021).

## 5. Results

Table 1 displays the descriptive data for the entire sample as well as the nutritional risk subgroups. Among 448 older individuals evaluated, a higher proportion of women (70.76%) was observed, as well as those aged between 60 and 69 years (45.31%), with four years or less of education (66.10%) and living with one person (43.41%). Their family income ranged between one and two minimum wages (44.31%). Regarding health conditions, most of the sample was independent in basic activities of daily living (87.04%) and instrumental activities of daily living (64.50%). They did not exhibit cognitive decline (70.52%) or depressive symptoms (65.24%). Concerning the presence of diseases, 69.82% of the older people reported having two or more chronic diseases (multimorbidity), 75.11% did not have cancer, and 41.57% were continuously using five or more medications per day (polypharmacy) (Table 1).

In addition, it was observed that 34.15% of the older individuals were classified as at risk of malnutrition/malnourished. There were differences regarding the risk of malnutrition in relation to years of education (p = 0.045), family income (p = 0.018), multimorbidity (p<0.001), cancer (p = 0.004), polypharmacy (p<0.001), basic activities of daily living (p = 0.023), instrumental activities of daily living (p<0.001), cognitive decline (p<0.001), depressive symptoms (p<0.001), and shorter telomere length (p = 0.022).

Regarding nutritional aspects, it was observed that most older people did not report a decrease in food intake (68.18%). They consumed a minimum of three meals per day (78.57%) and at least one daily serving of milk or dairy products, two or more weekly servings of vegetables or eggs, and meat, fish, or poultry daily (51.02%). The consumption of two or more daily servings of fruits or vegetables was reported by 76.31% of the participants, and 54.46% reported consuming more than five cups of fluids per day (Table 2).

Table 2 also show that all of participants lived in their own homes, 97.77% had normal mobility), 82% did not have skin lesions or bedsores and 58.26% used more than three medications per day. It was also observed that 58.64% of participants considered their own health to be better compared to their peers, 72.95% did not experience psychological stress or acute illness in the past 3 months, and 49.33% did not suffer from psychological problems.

**Table 1. Characterization and percentage distribution of older participants according to total sample and nutritional risk subgroups for all variables.**

| | Total n (%) | Nutritional risk | | $p^d$ |
|---|---|---|---|---|
| | | No n (%) | Yes n (%) | |
| **Sociodemographic** | | | | |
| **Sex** | | | | |
| Male | 131(29,24) | 91(69,47) | 40(30,53) | 0,299 |
| Female | 317(70,76) | 204(64,35) | 113(35,65) | |
| **Age groups** | | | | |
| 60 to 69 | 203(45,31) | 141(69,46) | 62 (30,54) | 0,238 |
| 70 to 79 | 172(38,39) | 111(64,53) | 61 (35,47) | |
| 80 and over | 73(16,30) | 43(58,90) | 30 (41,10) | |
| **Education** | | | | |
| > 4 years | 140(33,90) | 102(72,86) | 38(27,14) | 0,045 |
| ≤ 4 years | 273(66,10) | 172(63,00) | 101(37,00) | |
| **Household size** | | | | |
| Living alone | 81(18,41) | 54(66,67) | 27(33,33) | 0,094 |
| Lives with a one person | 191(43,41) | 118(61,78) | 73(38,22) | |
| Lives with two or more person | 168(38,18) | 122(72,62) | 46(27,38) | |
| **Familiar (wage) income** | | | | |
| > 2 minimum wage[a] | 144(34,87) | 106(73,61) | 38(26,39) | 0,018 |
| > 1 e ≤ 2 minimum wage | 183(44,31) | 109(59,56) | 74(40,44) | |
| ≤ 1 minimum wage | 86(20,82) | 61(70,93) | 25(29,07) | |
| **General Health/illness status** | | | | |
| **Multimorbidity** | | | | |
| No | 131(30,18) | 108(82,44) | 23(17,56) | <0,001 |
| Yes | 303(69,82) | 183(60,40) | 120(39,60) | |
| **Cancer** | | | | |
| No | 332(75,11) | 234(70,48) | 98(29,52) | 0,004 |
| Yes | 110(24,89) | 61(55,45) | 49(44,55) | |
| **Polypharmacy** | | | | |
| No | 253(58,43) | 188(74,31) | 65(25,69) | <0,001 |
| Yes | 180(41,57) | 106(58,89) | 74(41,11) | |
| **Functionality** | | | | |
| **BADL[b]** | | | | |
| Independent | 376(87,04) | 259(68,88) | 117(31,12) | 0,023 |
| Dependent | 56(12,96) | 30(53,57) | 26(46,43) | |
| **IADL[c]** | | | | |
| Independent | 278(64,50) | 205(73,74) | 73(26,26) | <0,001 |
| Dependent | 153(35,50) | 84(54,90) | 69(45,10) | |
| **Mental Health status** | | | | |
| **Cognitive profile** | | | | |
| No | 311(70,52) | 227(72,99) | 84(27,01) | <0,001 |
| Yes | 130(29,48) | 68(52,31) | 62(47,69) | |
| **Depressive symptoms** | | | | |
| No | 289(65,24) | 213(73,70) | 76(26,30) | <0,001 |
| Yes | 154(34,76) | 82(53,25) | 72(46,75) | |
| **Shorter telomere length** | | | | |
| No | 224(50,00) | 159(70,98) | 65(29,02) | 0,022 |

(*Continued*)

**Table 1.** (Continued)

| | Total n (%) | Nutritional risk | | $p^d$ |
| --- | --- | --- | --- | --- |
| | | No n (%) | Yes n (%) | |
| **Sociodemographic** | | | | |
| Yes | 224(50,00) | 136(60,71) | 88(39,29) | |

*Notes:* [a] Current minimum wage = R$998.00 (approximately US$250.00)

[b] ADL (Activities of Daily Living)

[c] IADL (Instrumental Activities of Daily Living)

[d] Chi-squared Test.

The results of the univariate analysis are presented in Table 3. All variables introduced in the model were statistical associated with the risk of malnutrition except the sociodemographic variables of sex, age groups and familiar (wage) income.

In Fig 2, AUROC represents the quality of the final model, indicating that the significant characteristics were able to explain 71% of the risk of malnutrition.

Table 4 displays the univariate correlations, accounting for potential confounding variables. In the final model, older individuals with shorter telomere length had a higher likelihood of having a risk of malnutrition (OR = 1.63; 95% CI = 1.04–2.55) compared to older individuals with longer telomeres, and regardless of age group, family income, multimorbidity, cognitive decline, and depressive symptoms.

## 6. Discussion

Our study delved into the intriguing interplay between telomere length and the risk of malnutrition/undernutrition among older adults. We sought to address a critical research question: Do older adults with shorter telomere lengths have an elevated susceptibility to malnutrition? The compelling results we have uncovered in this study bolster the existence of a noteworthy association between shorter telomere lengths and an augmented risk of malnutrition/undernutrition within the older population.

One of the most noteworthy findings from our investigation is the robustness of this relationship. Regardless of age range, family income, multimorbidity, cognitive decline, or depressive symptoms, individuals with shorter telomeres consistently exhibited a significantly higher likelihood of being at risk of malnutrition/undernutrition compared to their counterparts with longer telomeres. This robust correlation underscores the potential significance of telomere length as a marker for assessing nutritional vulnerability among the older adults.

### 6.1 Telomer lengths and nutrition associations

Our study contributes to the growing body of research in the field of gerontology, shedding light on an aspect that has not been extensively explored in prior literature. While the relationship between telomere length and aging has garnered considerable attention, the connection between telomere length and nutritional status among older adults remains a relatively uncharted territory.

To the best of our knowledge, a poor number of similar studies have been found that investigate the association between telomere length and the risk of malnutrition/undernutrition in older individuals. However, some studies have examined the effects of food deprivation during gestation and early life, which have implications for adult life and aging [32, 34]. Furthermore, our findings align with research which highlights the impact of socio-economic disparities on

**Table 2. Characterization and percentage distribution of older participants according to total sample and nutritional risk subgroups for all questions of mini nutritional assessment.**

| | Total n(%) | Nutritional risk | | P |
|---|---|---|---|---|
| | | No | Yes | |
| | | n (%) | n (%) | |
| **Dietary Intake** | | | | |
| Severe Decrease | 27(6,14) | 5(18,52) | 22(81,48) | <0,001[e] |
| Moderate Decrease | 113(25,68) | 54(47,79) | 59(52,21) | |
| No Decrease | 300(68,18) | 235(78,33) | 65(21,67) | |
| **Weight loss in the last 3 Months** | | | | |
| More than 3 kilograms | 48(10,91) | 8(16,67) | 40(83,33) | <0,001[d] |
| Cannot tell | 17(3,86) | 6(35,29) | 11(64,71) | |
| Between one and 3 kilograms | 77(17,50) | 41(53,25) | 36(46,75) | |
| No Weight Loss | 298(67,73) | 240(80,54) | 58(19,46) | |
| **Mobility** | | | | |
| Confined to bed or wheelchair | - | - | - | 0,081[e] |
| Ambulates within the home | 10(2,23) | 4(40,00) | 6(60,00) | |
| Normal | 438(97,77) | 291(66,44) | 147(33,56) | |
| **Psychological stress or acute illness in the last 3 months** | 119(27,05) | 42(35,29) | 77(64,71) | <0,001[d] |
| Yes | 321(72,95) | 252(78,50) | 69(21,50) | |
| No | 27(6,14) | 5(18,52) | 22(81,48) | <0,001[e] |
| **BMI[a]** | | | | |
| < 19 | 16(3,57) | 2(12,50) | 14(87,50) | <0,001[e] |
| 19 ≤ BMI < 21 | 20(4,46) | 7(35,00) | 13(65,00) | |
| 21 ≤ BMI < 23 | 40(8,93) | 27(67,50) | 13(32,50) | |
| ≥ 23 | 372(83,04) | 259(69,62) | 113(30,38) | |
| **Living in own home** | | | | |
| No | - | - | - | - |
| Yes | 448(100,00) | 295(65,85) | 153(34,15) | |
| **Uses more than 3 medications** | | | | |
| Yes | 261(58,26) | 149(57,09) | 112(42,91) | <0,001[d] |
| No | 187(41,74) | 146(78,07) | 41(21,93) | |
| **Skin lesion or bed sores** | | | | |
| Yes | 79(18,00) | 39(49,37) | 40(50,63) | <0,001[d] |
| No | 360(82,00) | 255(70,83) | 105(29,17) | |
| **Number of meals per day** | | | | |
| One meal | 6(1,34) | 1(16,67) | 5(83,33) | 0,024[e] |
| Two meals | 90(20,09) | 56(62,22) | 34(37,78) | |
| Three meals | 352(78,57) | 238(67,61) | 114(32,39) | |
| **Consumes at least: one daily serving of dairy or derivatives; two or more weekly servings of vegetables or eggs; meat, fish, or chicken daily** | | | | |
| 0 or 1 "yes" | 59(13,38) | 27(45,76) | 32(54,24) | <0,001[d] |
| 2 "yes" | 157(35,60) | 91(57,96) | 66(42,04) | |
| 3 "yes" | 225(51,02) | 177(78,67) | 48(21,33) | |
| **Consumes two or more daily servings of fruits or vegetables** | | | | |
| No | 104(23,69) | 56(53,85) | 48(46,15) | <0,001[d] |
| Yes | 335(76,31) | 238(71,04) | 97(28,96) | |
| **Daily liquid intake** | | | | |
| Less than three cups | 39(8,71) | 16(41,03) | 23(58,97) | 0,002[d] |
| Three to five cups | 165(36,83) | 108(65,45) | 57(34,55) | |
| More than five cups | 244(54,46) | 171(70,08) | 73(29,92) | |

*(Continued)*

**Table 2.** (Continued)

| | Total n(%) | Nutritional risk | | P |
|---|---|---|---|---|
| | | No | Yes | |
| | | n (%) | n (%) | |
| **Eating habits** | | | | |
| Needs assistance | 3(0,69) | 0(0,00) | 3(100,00) | 0,014[e] |
| Alone with difficulty | 9(2,06) | 4(44,44) | 5(55,56) | |
| Alone without difficulty | 424(97,25) | 290(68,40) | 134(31,60) | |
| **Believes to have a nutritional problem** | | | | |
| Believes to be malnourished | 37(8,45) | 6(16,22) | 31(83,78) | <0,001[e] |
| Cannot tell | 8(1,83) | 4(50,00) | 4(50,00) | |
| Believes not to have a nutritional problem | 393(89,72) | 284(72,26) | 109(27,74) | |
| **Comparison of the current state of health with other individuals of similar age.** | | | | |
| Not very good | 29(6,59) | 6(20,69) | 23(79,31) | <0,001[e] |
| Cannot tell | 11(2,50) | 5(45,45) | 6(54,55) | |
| Good | 142(32,27) | 89(62,68) | 53(37,32) | |
| Better | 258(58,64) | 194(75,19) | 64(24,81) | |
| **AC (cm)** | | | | |
| CB < 21 | 2(0,45) | 1(50,00) | 1(50,00) | 0,005[e] |
| 21 ≤ CB ≤ 22 | 8(1,79) | 1(12,50) | 7(87,50) | |
| CB > 22 | 438(97,76) | 293(66,89) | 145(33,11) | |
| **CC (cm)** | | | | |
| CP < 31 | 40(8,93) | 15(37,50) | 25(62,50) | <0,001[d] |
| CP ≥ 31 | 408(91,07) | 280(68,63) | 128(31,37) | |
| **Neuropsychological Problems** | | | | |
| Dementia or depression | 97(21,65) | 38(39,18) | 59(60,82) | <0,001[d] |
| Mild dementia | 130(29,02) | 68(52,31) | 62(47,69) | |
| No psychological problems | 221(49,33) | 189(85,52) | 32(14,48) | |

*Notes:* [a]BMI (Body Mass Index)

[b]AC (Arm Circumference)

[c]CC (Calf Circumference)

[d]Chi-squared Test

[e]Fisher's Exact Test.

both telomere length and nutritional status in older adults [60]. Their work underscores the multifaceted nature of these relationships and the importance of considering socio-economic factors in gerontological research.

To contextualize our findings, it is essential to consider related research on the broader implications of early-life experiences and environmental factors on aging and health outcomes. Previous research provides insights into the long-lasting consequences of early-life nutritional deficits on aging-related processes, including telomere attrition [61–63]. Their study underscores the potential role of early-life nutrition in shaping telomere length and, by extension, the risk of age-related health conditions. In this sense, it would be interesting to explore more deeply whether nutritional intervention strategies during critical developmental phases can attenuate this effect.

A study conducted in Amsterdam with 2.414 individuals of both sexes (mean age of 68 years old) concluded that leukocyte telomere length and the percentage of short telomeres did not differ between those exposed to famine during early gestation and those not exposed

**Table 3. Univariate analysis of the association between risk of malnutrition and all variables (n = 448).**

|  | OR[d] | *p* | 95% CI[e] |
|---|---|---|---|
| **Sociodemographic** |  |  |  |
| **Sex** |  |  |  |
| Male | 1,00 |  |  |
| Female | 1,26 | 0,300 | 0,81–1,95 |
| **Age groups** |  |  |  |
| 60 to 69 | 1,00 |  |  |
| 70 to 79 | 1,25 | 0,312 | 0,81–1,92 |
| 80 and over | 1,59 | 0,102 | 0,91–2,76 |
| **Education** |  |  |  |
| > 4 years | 1,00 |  |  |
| ≤ 4 years | 1,58 | 0,046 | 1,00–2,46 |
| **Household size** |  |  |  |
| Living alone | 1,00 |  |  |
| Lives with a one person | 1,24 | 0,445 | 0,72–2,14 |
| Lives with two or more person | 0,75 | 0,334 | 0,42–1,34 |
| **Familiar (wage) income** |  |  |  |
| > 2 minimum wage[a] | 1,00 |  |  |
| > 1 e ≤ 2 minimum wage | 1,89 | 0,008 | 1,18–3,04 |
| ≤ 1 minimum wage | 1,14 | 0,659 | 0,63–2,07 |
| **General Health/illness status** |  |  |  |
| **Multimorbidity** |  |  |  |
| No | 1,00 |  |  |
| Yes | 1,75 | <0,001 | 1,36–2,26 |
| **Cancer** |  |  |  |
| No | 1,00 |  |  |
| Yes | 1,92 | 0,004 | 1,23–2,99 |
| **Polypharmacy** |  |  |  |
| No | 1,00 |  |  |
| Yes | 2,02 | 0,001 | 1,34–3,04 |
| **Functionality** |  |  |  |
| **BADL[b]** |  |  |  |
| Independent | 1,00 |  |  |
| Dependent | 1,92 | 0,025 | 1,09–3,39 |
| **IADL[c]** |  |  |  |
| Independent | 1,00 |  |  |
| Dependent | 2,30 | <0,001 | 1,52–3,49 |
| **Mental Health status** |  |  |  |
| **Cognitive profile** |  |  |  |
| No | 1,00 |  |  |
| Yes | 2,46 | <0,001 | 1,61–3,77 |
| **Depressive symptoms** |  |  |  |
| No | 1,00 |  |  |
| Yes | 2,46 | <0,001 | 1,63–3,71 |
| **Shorter telomere length** |  |  |  |
| No | 1,00 |  |  |

(*Continued*)

**Table 3.** (Continued)

| | OR[d] | *p* | 95% CI[e] |
|---|---|---|---|
| **Sociodemographic** | | | |
| Yes | 1,58 | 0,022 | 1,07–2,35 |

*Notes:* [a]Current minimum wage = R$998.00 (approximately US$250.00)

[b]ADL (Activities of Daily Living)

[c]IADL (Instrumental Activities of Daily Living)

[d]Odds Ratio

[e]95% CI (95% Confidence Interval).

during gestation. The variables associated with shorter telomere length were low socioeconomic status, frequent alcohol consumption, history of cancer, and poorer overall health [32]. Considering the findings of our study, one could speculate that common factors may play crucial roles in both telomere shortening and vulnerability to malnutrition in older populations. While the Amsterdam study did not find differences in telomere length concerning exposure to early gestational famine, our results suggest that these socioeconomic and health-related factors may be important considerations when assessing nutritional risk in older individuals.

Another study conducted in St. Petersburg with 356 individuals (aged 64 to 82 years) evaluated cardiovascular health, cardiovascular aging markers, and telomere length in survivors of the Siege of Leningrad. They found that early-life famine, especially during intrauterine and

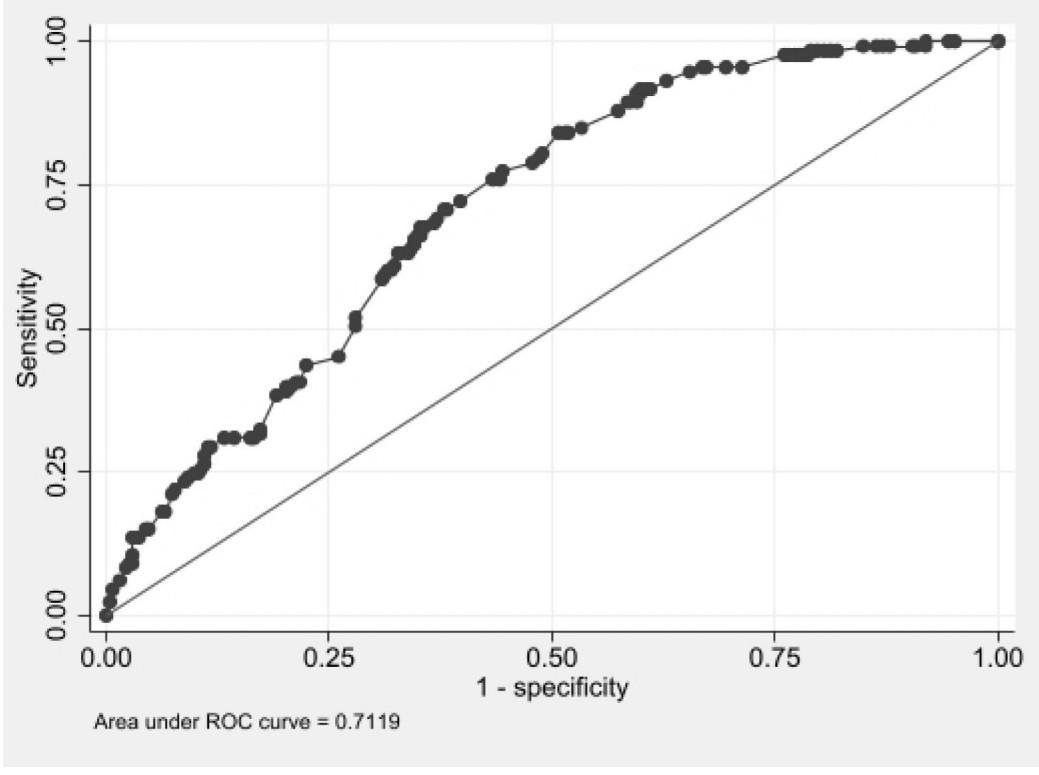

**Fig 2. Area under ROC curve representing the fit of final model of logistic regression.**

**Table 4. Multivariate analysis of the association between risk of malnutrition and shorter telomere length (n = 448).**

| | OR[b] | p | 95% CI[c] |
|---|---|---|---|
| **Age groups** | | | |
| 60 to 69 | 1,00 | | |
| 70 to 79 | 1,08 | 0,770 | 0,65–1,79 |
| 80 and over | 1,36 | 0,382 | 0,68–2,69 |
| **Familiar (wage) income** | | | |
| > 2 minimum wage[a] | 1,00 | | |
| > 1 e ≤ 2 minimum wage | 1,85 | 0,019 | 1,10–3,11 |
| ≤ 1 minimum wage | 1,04 | 0,908 | 0,55–1,97 |
| **Multimorbidity** | | | |
| No | 1,00 | | |
| Yes | 1,64 | 0,001 | 1,24–2,17 |
| **Cognitive profile** | | | |
| No | 1,00 | | |
| Yes | 2,07 | 0,006 | 1,23–3,47 |
| **Depressive symptoms** | | | |
| No | 1,00 | | |
| Yes | 1,95 | 0,005 | 1,22–3,13 |
| **Shorter telomere length** | | | |
| No | 1,00 | | |
| Yes | 1,63 | 0,034 | 1,04–2,55 |

*Notes:* [a]Current minimum wage = R$998.00 (approximately US$250.00)

[b]Odds Ratio

[c]95% CI (95% Confidence Interval). Hosmer–Lemeshow test: p = 0.344

late childhood periods, may contribute to accelerated aging with telomere shortening in both sexes, but it does not have a direct effect on the prevalence of cardiovascular diseases and risk factors after seven decades of exposure [34]. In line with our study's results, we could speculate that these findings offer insights into potential mechanisms behind this association. While the St. Petersburg study didn't directly link childhood famine to increased cardiovascular disease prevalence after seven decades, it implies a complex interplay between early nutritional deprivation, telomere dynamics, and long-term health effects warranting further investigation.

Building upon previous research supporting the connection between food restriction and its potential impact on epigenetic changes, gene expression, and cellular aging, as well as the influence of environmental factors during pregnancy and early postnatal life on this interaction [64], it is intriguing to speculate how these modifications, including variations in telomere length, may have enduring implications for cellular aging, in line with the findings of our study. Additionally, it is worth noting a recent study which emphasized the influence of maternal nutrition on determining telomere length in offspring, further underscoring the complexity of this connection and the importance of future investigations to elucidate these mechanisms [65].

## 6.2 Strengths and limitations

This study has some limitations that should be considered. Firstly, the cross-sectional design restricts establishing causality or temporal sequence. While it provides valuable insights into the

association between telomere length and malnutrition risk in older adults, it cannot definitively determine if shorter telomeres directly contribute to higher risk or if other factors are involved. Future longitudinal studies are needed for causal elucidation. Secondly, excluding older adults with mobility impairments may introduce selection bias and limit generalizability. Mobility impairments are known as a malnutrition risk factor, and their exclusion may underestimate the true prevalence of malnutrition risk. Including a more diverse sample, encompassing individuals with mobility impairments, would provide a comprehensive understanding of the relationship between telomere length, malnutrition risk, and mobility status.

Despite its limitations, this study possesses significant strengths that enhance its validity and significance. Firstly, it contributes to the existing literature by advancing our understanding of telomeres as potential biomarkers of aging-related health outcomes, specifically in relation to nutritional status in older adults. Secondly, the study benefits from a large sample size, increasing the generalizability of the findings and allowing for robust statistical analyses. Furthermore, the comprehensive analysis approach considers multiple factors associated with malnutrition risk, providing a nuanced understanding of the complex interactions between these variables and their impact on nutritional status.

## 6.3 Practical applications

The findings of this research have significant practical applications for addressing malnutrition in older adults, aligning with the United Nations Sustainable Development Goal 3, which aims to ensure healthy lives and promote well-being for all at all ages. By identifying the risk factors associated with malnutrition in older adults, this study provides valuable insights for healthcare professionals, especially nutritionists and dietitians, in developing targeted interventions and strategies for prevention and treatment. For example, healthcare professionals can conduct more frequent and detailed nutritional assessments and implement personalized nutritional intervention plans. Additionally, rehabilitation programs for the older adults could include specific strategies to improve nutrition, focusing on individuals at higher risk. In terms of prevention, nutritional education and awareness initiatives could be directed towards this population. Such approaches can help mitigate the risks of malnutrition and its consequences, promoting better quality of life and health outcomes for older adults.

## 6.4 Future directions for futures studies

By demonstrating the association between shorter telomere length and the risk of malnutrition/malnourishment, it can contribute to the scientific community via strengthening the hypothesis that telomere length may be a biomarker of aging, as it is associated with adverse outcomes. Investigating the mechanisms through which telomere length affects nutritional status and exploring the potential role of telomere length as a biomarker for aging-related health outcomes could provide a deeper understanding of the underlying biological processes involved. Future studies should focus on conducting longitudinal research to understand the temporal relationship between genetic information and the development of malnutrition in older adults.

## 7. Conclusion

Older individuals characterized by shorter telomere length exhibit a higher susceptibility to malnutrition and malnutrition risk compared to those with longer telomeres, independent of age group, family income, multimorbidity, cognitive decline, and depressive symptoms. Given the escalating number of older individuals and the detrimental impact of malnutrition and malnutrition risk on this population, early detection of these nutritional disorders enables

timely interventions, thus preserving a higher quality of life and overall health among this cohort. The utilization of telomere length as a biomarker of aging, specifically in conjunction with nutritional status assessment, holds promise in fostering the advancement of novel clinical trials. This approach facilitates the implementation of targeted interventions aimed at preventing and treating the risk of malnutrition. This integration holds the potential to yield substantial benefits for older individuals, fostering their well-being and overall health.

## Author Contributions

**Conceptualization:** Priscila Rodrigues, Daniela Lima, Tábatta Brito.

**Data curation:** Ricardo Vieira, Ariene Orlandi, Daniela Lima, Tábatta Brito.

**Formal analysis:** Priscila Rodrigues, Ricardo Vieira, Daniela Lima, Tábatta Brito.

**Funding acquisition:** Sónia Brito-Costa, Ana Moisão, Daniela Lima, Tábatta Brito.

**Investigation:** Priscila Rodrigues, Guilherme Furtado, Ligiana Corona, Daniela Lima, Tábatta Brito.

**Methodology:** Priscila Rodrigues, Ricardo Vieira, Ariene Orlandi, Ligiana Corona, Daniela Lima, Tábatta Brito.

**Resources:** Guilherme Furtado, Margarida Martins, Ariene Orlandi, Sónia Brito-Costa, Ligiana Corona.

**Software:** Sónia Brito-Costa.

**Supervision:** Guilherme Furtado, Sónia Brito-Costa.

**Validation:** Guilherme Furtado, Margarida Martins, Sónia Brito-Costa, Ana Moisão, Tábatta Brito.

**Visualization:** Guilherme Furtado, Sónia Brito-Costa, Ana Moisão.

**Writing – original draft:** Guilherme Furtado, Sónia Brito-Costa, Daniela Lima, Tábatta Brito.

**Writing – review & editing:** Guilherme Furtado, Margarida Martins, Sónia Brito-Costa, Tábatta Brito.

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
