## [Decision Letter · Decision Letter 0]

2 Jul 2024

PONE-D-23-39724Exposing Telomere Length's Impact on Malnutrition Risk among Older Adults Residing in the Community: Insights from Cross-Sectional Data AnalysisPLOS ONE

Dear Dr. Brito-Costa,

Thank you for submitting your manuscript to PLOS ONE. After careful consideration, we feel that it has merit but does not fully meet PLOS ONE’s publication criteria as it currently stands. Therefore, we invite you to submit a revised version of the manuscript that addresses the points raised during the review process.

Please address all the comments from the reviewer thoroughly

We look forward to receiving your revised manuscript.

Kind regards,

Gabriele Saretzki, PhD

Academic Editor

PLOS ONE

Journal Requirements:

"This article was supported by the National Council for Scientific and Technological Development (CNPq), Grant No. 429823/2018-5-MCTIC/CNPq No. 28/2018), and by Fundação de Amparo à Pesquisa do Estado de Minas Gerais—FAPEMIG (Grant No. APQ-01168-18; 001/2018)."

**Additional Editor Comments:**

Please follow the reviewers suggestions and improve the manuscript.

Reviewers' comments:

Reviewer's Responses to Questions

**Comments to the Author**

1. Is the manuscript technically sound, and do the data support the conclusions?

Reviewer #1: Yes

2. Has the statistical analysis been performed appropriately and rigorously? 

Reviewer #1: Yes

3. Have the authors made all data underlying the findings in their manuscript fully available?

Reviewer #1: No

4. Is the manuscript presented in an intelligible fashion and written in standard English?

Reviewer #1: Yes

5. Review Comments to the Author

Reviewer #1: The manuscript presents an important study on the association between telomere length and the risk of malnutrition among older adults. The research is methodologically sound, the data analysis is robust, and the findings are significant. The manuscript is well-structured and clearly written, making it a valuable contribution to the field of geriatric science.

Specific Comments:

1. Title and Abstract:

* The title is informative and accurately reflects the study's content.

* The abstract provides a concise and clear summary of the study. It effectively highlights the background, objectives, methods, results, and conclusions.

2. Introduction:

* The introduction is comprehensive and provides a good context for the study. It outlines the importance of investigating the relationship between telomere length and nutritional status in older adults.

* The literature review is thorough and identifies the gap that this study aims to fill.

3. Methods:

* The study design is appropriate for the research question. The cross-sectional design and the probabilistic sampling method are well-explained.

* The data collection process, including the use of validated tools like the Mini Nutritional Assessment and real-time qPCR for telomere length measurement, is clearly described.

* The statistical methods are appropriate and well-justified. However, providing more detail on the rationale for choosing specific statistical tests would enhance clarity. The software used for statistical analysis, specifically for the regression model, should be indicated. Currently, only the software for sample size calculation (G*Power) is mentioned.

* Suggestion for Additional Analysis:

* Consider including the following parameters to provide a more comprehensive evaluation of the model's fit (e.g. -2 Log Likelihood, Pseudo R^2 (Nagelkerke, Cox & Snell), Model Chi-Square, and a parameter to evaluate the model fit, maybe Hosmer-Lemeshow Test?)

* Additionally, only if the authors deemed interesting, adding ROC Curve and AUC to provide information on the discriminative power of the model could be valuable.

4. Results:

* The results are presented clearly and concisely. Tables and figures are used effectively to summarize the data.

* The findings indicate a significant association between shorter telomere length and increased risk of malnutrition. These results are discussed in the context of existing literature.

5. Discussion:

* The discussion section is well-organized and provides a comprehensive interpretation of the findings.

* The authors acknowledge the limitations of the cross-sectional design and potential selection bias. They also highlight the study's strengths, such as the large sample size and comprehensive analysis.

* The practical implications of the findings are discussed, and suggestions for future research are provided.

6. Conclusion:

* The conclusion succinctly summarizes the study's contributions and emphasizes the potential of telomere length as a biomarker for assessing nutritional risk in older adults.

* The call for further research to explore the underlying mechanisms and conduct longitudinal studies is well-founded.

7. Ethical Considerations:

* The manuscript includes an appropriate ethics statement, indicating approval from the relevant ethics committee and informed consent from participants.

8. Funding and Acknowledgments:

* The sources of funding are clearly disclosed, and the authors have declared no competing interests.

* The acknowledgments section appropriately credits those who contributed to the study.

9. Data Availability Statement:

* Suggestion: The manuscript currently does not include a data availability statement. PLOS ONE requires authors to make all data underlying the findings described in their manuscript fully available without restriction. Please include a data availability statement describing where the data supporting the findings can be accessed. This should be part of the manuscript or its supporting information, or deposited in a public repository. If there are any restrictions on publicly sharing data, such as participant privacy or use of data from a third party, please specify these restrictions.

Recommendations for Revision:

1. Statistical Methods:

* Provide more detail on the rationale for choosing specific statistical tests in the methods section.

* Indicate the software used for the statistical analysis, specifically for the regression model.

2. Practical Applications:

* Include more specific examples of how the findings can be applied in clinical settings to prevent and treat malnutrition in older adults.

3. Additional Analysis:

* Incorporate the model evaluation parameters to provide a more comprehensive evaluation of the model's fit.

* Additionally, if deemed interesting, adding ROC Curve and AUC to provide information on the discriminative power of the model could be valuable.

4. Add Data Availability Statement:

* Include a data availability statement as per PLOS ONE's requirements. Describe where the data supporting the findings can be accessed, either as part of the manuscript, its supporting information, or deposited in a public repository. Specify any restrictions on data sharing, if applicable.

Conclusion: This manuscript provides valuable insights into the relationship between telomere length and malnutrition risk in older adults. It is methodologically sound, well-written, and makes a significant contribution to the field. I recommend it for publication with minor revisions to enhance clarity and ensure the results are communicated as effectively as possible.

6. PLOS authors have the option to publish the peer review history of their article (what does this mean?). If published, this will include your full peer review and any attached files.

Reviewer #1: No

---

## [Author Response · Author response to Decision Letter 0]

18 Jul 2024

Response to reviewer’s:

Reviewer’s requests:

1. Statistical Methods: 

* Provide more detail on the rationale for choosing specific statistical tests in the methods section.

Response: Thank you very much. The information related to the specific statistical tests is now inserted on the methods section point 4 (Statistical analysis) first paragraph

* Indicate the software used for the statistical analysis, specifically for the regression model.

Response: Thank you very much. The information related to the regression model’s used software is now inserted on the methods section point 4 (Statistical analysis) last paragraphs.

2. Practical Applications:

* Include more specific examples of how the findings can be applied in clinical settings to prevent and treat malnutrition in older adults. 

Response: Thank you very much for your advice. We have reformulated the point 6.3 Practical applications.

3. Additional Analysis: 

* Incorporate the model evaluation parameters to provide a more comprehensive evaluation of the model's fit.

* Additionally, if deemed interesting, adding ROC Curve and AUC to provide information on the discriminative power of the model could be valuable.

Response: Thank you for your advice. We have inserted the following text in results section: In Figure 2, AUROC represents the quality of the final model, indicating that the significant characteristics were able to explain 71% of the risk of malnutrition.

We also have inserted figure 2 (Area under ROC curve representing the fit of final model of logistic regression)before table 4

We also inserted a new note in table 4: Hosmer–Lemeshow test: p=0.344

4. Add Data Availability Statement:

* Include a data availability statement as per PLOS ONE's requirements. Describe where the data supporting the findings can be accessed, either as part of the manuscript, its supporting information, or deposited in a public repository. Specify any restrictions on data sharing, if applicable.

Response: Thank you very much. We have inserted Data Statement information: data will be available if requested to the corresponding author:

We would also like to emphasize that we have taken the freedom to replace the term "elderly" with the word "older people" throughout the manuscript to respect the new guidelines of the American Medical Association, the American Psychological Association, Associated Press and the Gerontological Society of America with regard to the elimination of ageist language and to reduce negative stereotypes about older people. https://publichealth.wustl.edu/age-inclusive-language-are-you-using-it-in-your-writing-and-everyday-speech/#:~:text=Terms%20like%20seniors%2C%20elderly%2C%20the,the%20older%20population%20are%20preferred.

---

## [Editor Report · Decision Letter 1]

29 Jul 2024

Exposing Telomere Length's Impact on Malnutrition Risk among Older Adults Residing in the Community: Insights from Cross-Sectional Data Analysis

PONE-D-23-39724R1

Dear Dr. Brito-Costa,

We’re pleased to inform you that your manuscript has been judged scientifically suitable for publication and will be formally accepted for publication once it meets all outstanding technical requirements.

Kind regards,

Gabriele Saretzki, PhD

Academic Editor

PLOS ONE

Additional Editor Comments (optional):

the authors have carefully and comprehensively addressed the minor comments from the reviewer which greatly improved the manuscript,
---

## [Editor Report · Acceptance letter]

8 Aug 2024

PONE-D-23-39724R1 

PLOS ONE

Dear Dr. Brito-Costa, 

I'm pleased to inform you that your manuscript has been deemed suitable for publication in PLOS ONE. Congratulations! Your manuscript is now being handed over to our production team.

Kind regards, 

on behalf of

Dr. Gabriele Saretzki 

Academic Editor

PLOS ONE